# Study protocol: A systematic review and meta-analysis regarding the influence of coagulopathy and immune activation on new onset atrial fibrillation in patients with sepsis

Michael Downes[1]*, Ingeborg D. Welters[1,2,3], Brian W. Johnston[1,2,3]

1 School of Medicine, Faculty of Health and Life Sciences, Institute of Life Course and Medical Sciences, University of Liverpool, Liverpool, United Kingdom, 2 Liverpool Centre for Cardiovascular Science at University of Liverpool, Liverpool John Moores University and Liverpool Heart & Chest Hospital, Liverpool, United Kingdom, 3 Liverpool University Hospitals NHS Foundation Trust, Liverpool, United Kingdom

* hlmdowne@liverpool.ac.uk

## Abstract

### Background

New onset atrial fibrillation (NOAF) is the most common arrhythmia affecting critically ill patients with sepsis. NOAF is associated with increased intensive care unit mortality, increased hospital mortality, development of heart failure and increased risk of permanent atrial fibrillation and thromboembolic events such as stroke. The pathophysiology of NOAF has been outlined, however, a knowledge gap exists regarding the association between abnormalities in coagulation and immune biomarkers, and the risk of developing NOAF in patients with sepsis.

### Methods and analysis

This protocol describes a systematic review and meta-analysis following the Preferred Reporting Items for Systematic Review and Meta-Analysis Protocols guideline (PRISMA-P) and the Meta-Analyses and Systematic Reviews of Observational Studies guideline (MOOSE). We will conduct the literature search in Medline, Scopus and Cochrane Library. We will include studies that report data in adult patients (>18 years) with sepsis that develop NOAF. We will extract data from studies that report at least one coagulation or immune bio-marker. Risk of bias will be assessed by using the Newcastle Ottawa Scale (NOS) and Risk of Bias 2 tool (RoB2) for non-randomized and randomized trials respectively. The Grading of Recommendations Assessment, Development and Evaluation (GRADE) approach will be utilized in assessing the quality of evidence.

### Discussion

This systematic review and meta-analysis will explore the scientific literature regarding the association between coagulation and immune activation in critically ill patients with sepsis, who develop NOAF. The findings will add to the existing knowledge base of NOAF in sepsis,

**Data Availability Statement:** No datasets were generated or analysed during the current study. All

relevant data from this study will be made available upon study completion.

**Funding:** The author(s) received no specific funding for this work.

**Competing interests:** The authors have declared that no competing interests exist.

**Abbreviations:** NOAF, New onset atrial fibrillation; PRISMA-P, Preferred Reporting Items for Systematic Review and Meta-Analysis Protocols; MOOSE, Meta-Analyses and Systematic Reviews of Observational Studies; PROSPERO, International Prospective Register of Systematic Reviews; MeSH, Medical subject headings; NOS, Newcastle Ottawa Scale; RoB2, Risk of Bias 2; GRADE, Grading of Recommendations Assessment, Development and Evaluation.

highlight areas of uncertainty and identify future areas of interest to guide and improve management strategies for NOAF.

## Trial registration

**Registration details.** CRD42022385225 (PROSPERO).

## Introduction

Atrial fibrillation (AF) is the most common arrhythmia in patients admitted to the intensive care unit [1, 2]. New onset atrial fibrillation (NOAF) occurs in patients without a past medical history of AF and is particularly common in patients admitted with sepsis. The incidence of NOAF is estimated to be between 5–15% in all patients with sepsis [3–8] with some studies reporting even higher incidences of up to 40% in patients admitted with septic shock [7, 9, 10]. Risk factors for the development NOAF during admission to intensive care include: increasing age, male sex, pre-existing cardiovascular disease, acute renal failure, and acute respiratory failure, with sepsis itself conferring a moderate risk for NOAF [11, 12]. NOAF in patients with sepsis carries an increased risk of both in-hospital and ICU mortality [3, 13]. Further complications due to NOAF include increased risk of stroke, re-hospitalisation with AF, and the development of heart failure [13, 14].

In sepsis, interactions between the immune system and coagulation pathways result in clot formation, tissue damage and subsequent fibrosis of the myocardium [15, 16]. It is thought that accelerated atrial remodeling due to inflammation and infection promotes the development of NOAF in septic critically ill patients [1]. This atrial remodeling creates an arrhythmogenic substrate through which re-entrant electrical signals may occur resulting in NOAF [12]. NOAF in critical care is common and carries long-term clinical consequences for patients, therefore further research in this field is urgently required. Scientific evidence regarding the pathophysiology of NOAF in sepsis has been described, however, at present little is known about the association between coagulation and immune markers, and the risk of developing NOAF. Early identification of patients who are at risk of developing NOAF could assist clinicians in targeted interventions aimed at reducing its short and long-term consequences.

In the current protocol we describe a systematic review and meta-analysis to investigate the association between coagulopathy and immune activation in critically ill patients admitted with sepsis, who develop NOAF.

### Methods and analysis (research plan)

**Protocol design and registration.**   The study protocol for this systematic review will be conducted in accordance with the Preferred Reporting Items for Systematic Review and Meta-Analysis Protocols guideline (PRISMA-P) [17–20] (S1 Fig). In addition, any subsequent meta-analysis will follow the Meta-Analyses and Systematic Reviews of Observational Studies guideline (MOOSE) [21, 22].

The protocol has been registered with the International Prospective Register of Systematic Reviews (PROSPERO) CRD42022385225.

### Data sourcing and search strategy

The literature review will be undertaken in a systematic manner utilizing a predefined search strategy. Searches will be performed using three databases: Medline, Scopus and Cochrane

Library. A research librarian (AH) has participated in the development of this search strategy. Five key concepts were generated: sepsis, immune markers, coagulopathy, coagulation markers and atrial fibrillation. Lists of keywords were derived from the key concepts and utilized in all three databases to search for articles, with a title or abstract featuring at least one keyword.

In addition, medical subject heading (MeSH) terms will be used in Medline and Cochrane Library to assist in identifying further articles that would be of use. Boolean operators (OR, AND), truncation (*) for various suffixes and quotation marks ("") for phrases are to be utilized throughout the search strategy (Table 1).

Search results will be exported to EndNote X9 (Clarivate) and screened by two independent reviewers (MD, BWJ), a third reviewer (IW) will resolve any disagreements. There will be no limitations regarding the date of publication for the studies, ensuring any available articles are identified in the search.

## Outcome measures

### Primary outcome measure.

1. Development of NOAF in patients with sepsis.

### Secondary outcome measures.

1. Development of paroxysmal atrial fibrillation

2. Development of permanent atrial fibrillation

3. Mortality

4. Intensive care unit admission

5. Length of intensive care unit stay

6. Length of hospital stay

7. Development of acute renal failure

8. Need for renal replacement therapy

9. Need for vasopressor therapy

## Inclusion and exclusion criteria

Only human studies will be considered eligible for inclusion in this study, as a result both animal and in vitro studies will be excluded. This is because we wish to explore the association between coagulopathy, immune markers, and NOAF in patients with sepsis utilizing available studies performed in the hospital setting.

Participants under the age of 18 will be excluded, as septic pathophysiology in pediatric patients differs considerably from the response seen in adults [23, 24].

Our main outcome of interest is the development of NOAF, therefore we will include articles that report quantitative data on the incidence of NOAF as a complication of sepsis. We wish to determine if coagulation or inflammatory biomarkers can be used to predict NOAF; therefore we will include studies that report at least one biomarker of coagulation or immune activation in patients that develop NOAF in sepsis. Where possible we will include studies that report data in NOAF and non-NOAF populations.

Pharmaceutical trials will be excluded as they report data regarding sepsis and NOAF in the context of a particular drug and focus on outcomes regarding the success of treatment.

**Table 1. Complete search strategy and keywords (biomarkers).**

| Database | Search Strategy |
|---|---|
| Medline | 1. Sepsis/ (map term to subject heading)<br>2. (Sepsis, septic, septicemia, Toxemia, Toxaemia, Bacteraemia, Bacteremia) ab, ti.<br>3. 1 OR 2<br>4. Inflammation/ (map term to subject heading)<br>5. (Inflammation, inflamm*, CRP, "C-reactive protein", ESR, "Erythrocyte sedimentation rate", "plasma viscosity", "procalcitonin", PCT, IL-6, IL-1, "Tumour Necrosis Factor alpha", "Serum amyloid A", SAA, "heat shock proteins", HSPs, TNF, "Tumour Necrosis Factor", IL-10, "acute phase proteins", ferritin, "soluble IL-2 receptor", "white cell count", monocyte, basophil, lymphocyte, eosinophil, neutrophil, "neutrophil-lymphocyte ratio", "neutrophil lymphocyte ratio", "systemic inflammatory response index") ab, ti.<br>6. 4 OR 5<br>7. Disseminated Intravascular Coagulation/ (map term to subject heading)<br>8. Blood Coagulation/ (map term to subject heading)<br>9. (Coagulopathy, "sepsis induced coagulopath*", "SIC", "disseminated intravascular coagulation", "DIC", "sepsis associated coagulopath*", "SAC", "coagulation marker*", fibrinogen, prothrombin, "antithrombin III", beta-Thromboglobulin, "beta Thromboglobulin", "von Willebrand Factor*", Thrombin, "Blood Coagulation Factor*", "Factor V", calcium, "factor IV", "Factor VII", "Factor VIII", "Factor IX", "Factor X", "Factor XI", "Factor XII", "Factor XIII", fibrin, "antifibrinolytic agent*", "platelet count", "Mean Platelet Volume", P-Selectin, "P Selectin", thromboplastin, "protein C", "protein C inhibitor*", "protein S", thrombomodulin, "plasminogen inactivator*", "plasminogen activator*", plasminogen, carboxypeptidases, "thrombin-activatable fibrinolysis inhibitor", TAFI, "fibrinopeptide A", "fibrinopeptide B", hemostat*, haemostat*, thrombocyt*, platelet*, D-dimer*, "tissue factor*", thrombophilia, thrombocytopenia, PT, "prothrombin time", aPTT, "activated partial thromboplastin time", INR, "international normalised ratio", "APTT Ratio", "vitamin K", "fibrin degradation products", FDP, "fibrin split products" OR FSP, "Plasminogen activator inhibitor-1", PAI-1, "Alpha-2 plasmin inhibitor", "antiplasmin", thromboelasto*) ab, ti.<br>10. 7 OR 8 OR 9<br>11. (coagul* AND marker*) ab, ti.<br>12. 10 OR 11<br>13. Atrial Fibrillation/ (map term to subject heading)<br>14. ("atrial fibrillation", "AF", "new onset atrial fibrillation", new-onset-atrial-fibrillation, "NOAF", "atrial flutter*", "supraventricular tachycardia", supraventricular-tachycardia, "supraventricular arrhythmia", supraventricular-arrhythmia, "SVT", "atrial tachycardia") ab, ti.<br>15. 13 OR 14<br>16. 6 OR 12<br>17. 3 AND 15 AND 16 |
| Cochrane Library | 1. MeSH descriptor: [Sepsis] explode all trees<br>2. (sepsis OR septic OR septicemia OR Toxemia OR Toxaemia OR Bacteraemia OR Bacteremia):ti,ab,kw (Word variations have been searched)<br>3. 1 OR 2<br>4. MeSH descriptor: [Inflammation] explode all trees<br>5. (inflammation OR inflamm* OR CRP OR "C-reactive protein" OR ESR OR "Erythrocyte sedimentation rate" OR "plasma viscosity" OR "procalcitonin" OR PCT OR IL-6 OR IL-1 OR "Tumour Necrosis Factor alpha" OR "Serum amyloid A" OR SAA OR "heat shock proteins" OR HSPs OR TNF OR "Tumour Necrosis Factor" OR IL-10 OR "acute phase proteins" OR ferritin OR "soluble IL-2 receptor" OR "white cell count" OR monocyte OR basophil OR lymphocyte OR eosinophil OR neutrophil OR "neutrophil-lymphocyte ratio" OR "neutrophil lymphocyte ratio" OR "systemic inflammatory response index"):ti,ab,kw (Word variations have been searched)<br>6. MeSH descriptor: [Disseminated Intravascular Coagulation] explode all trees<br>7. MeSH descriptor: [Blood Coagulation] explode all trees<br>8. (coagulopathy OR "sepsis induced coagulopath*" OR "SIC" OR "disseminated intravascular coagulation" OR "DIC" OR "sepsis associated coagulopath*" OR "SAC" OR "coagulation marker*" OR fibrinogen OR prothrombin OR "antithrombin III" OR beta-Thromboglobulin OR "beta Thromboglobulin" OR "von Willebrand Factor*" OR Thrombin OR "Blood Coagulation Factor*" OR "Factor V" OR calcium OR "factor IV" OR "Factor VII" OR "Factor VIII" OR "Factor IX" OR "Factor X" OR "Factor XI" OR "Factor XII" OR "Factor XIII" OR fibrin OR "antifibrinolytic agent*" OR "platelet count" OR "Mean Platelet Volume" OR P-Selectin OR "P Selectin" OR thromboplastin OR "protein C" OR "protein C inhibitor*" OR "protein S" OR thrombomodulin OR "plasminogen inactivator*" OR "plasminogen activator*" OR plasminogen OR carboxypeptidases OR "thrombin-activatable fibrinolysis inhibitor" OR TAFI OR "fibrinopeptide A" OR "fibrinopeptide B" OR hemostat* OR haemostat* OR thrombocyt* OR platelet* OR D-dimer* OR "tissue factor*" OR thrombophilia OR thrombocytopenia OR PT OR "prothrombin time" OR aPTT OR "activated partial thromboplastin time" OR INR OR "international normalised ratio" OR "APTT Ratio" OR "vitamin K" OR "fibrin degradation products" OR FDP OR "fibrin split products" OR FSP OR "Plasminogen activator inhibitor-1" OR PAI-1 OR "Alpha-2 plasmin inhibitor" OR "antiplasmin" OR thromboelasto*):ti,ab,kw (Word variations have been searched)<br>9. 6 OR 7 OR 8<br>10. (coagul* AND marker*):ti,ab,kw (Word variations have been searched)<br>11. 9 OR 10<br>12. 4 OR 5<br>13. 11 OR 12<br>14. MeSH descriptor: [Atrial Fibrillation] explode all trees<br>15. ("atrial fibrillation" OR "AF" OR "new onset atrial fibrillation" OR new-onset-atrial-fibrillation OR "NOAF" OR "atrial flutter*" OR "supraventricular tachycardia" OR supraventricular-tachycardia OR "supraventricular arrhythmia" OR supraventricular-arrhythmia OR "SVT" OR "atrial tachycardia"):ti,ab,kw<br>16. 14 OR 15<br>17. 3 AND 13 AND 16 |

*(Continued)*

**Table 1.** (Continued)

| Database | Search Strategy |
|---|---|
| Scopus | 1. TITLE-ABS-KEY (sepsis OR septic OR septicemia OR toxemia OR toxaemia OR bacteraemia OR bacteremia)<br>2. TITLE-ABS-KEY (inflammation OR inflamm* OR crp OR "C-reactive protein" OR esr OR "Erythrocyte sedimentation rate" OR "plasma viscosity" OR "procalcitonin" OR pct OR il-6 OR il-1 OR "Tumour Necrosis Factor alpha" OR "Serum amyloid A" OR saa OR "heat shock proteins" OR hsps OR tnf OR "Tumour Necrosis Factor" OR il-10 OR "acute phase proteins" OR ferritin OR "soluble IL-2 receptor" OR "white cell count" OR monocyte OR basophil OR lymphocyte OR eosinophil OR neutrophil OR "neutrophil-lymphocyte ratio" OR "neutrophil lymphocyte ratio" OR "systemic inflammatory response index")<br>3. TITLE-ABS-KEY (coagulopathy OR "sepsis induced coagulopath*" OR "SIC" OR "disseminated intravascular coagulation" OR "DIC" OR "sepsis associated coagulopath*" OR "SAC" OR "coagulation marker*" OR fibrinogen OR prothrombin OR "antithrombin III" OR beta-thromboglobulin OR "beta Thromboglobulin" OR "von Willebrand Factor*" OR thrombin OR "Blood Coagulation Factor*" OR "Factor V" OR calcium OR "factor IV" OR "Factor VII" OR "Factor VIII" OR "Factor IX" OR "Factor X" OR "Factor XI" OR "Factor XII" OR "Factor XIII" OR fibrin OR "antifibrinolytic agent*" OR "platelet count" OR "Mean Platelet Volume" OR p-selectin OR "P Selectin" OR thromboplastin OR "protein C" OR "protein C inhibitor*" OR "protein S" OR thrombomodulin OR "plasminogen inactivator*" OR "plasminogen activator*" OR plasminogen OR carboxypeptidases OR "thrombin-activatable fibrinolysis inhibitor" OR tafi OR "fibrinopeptide A" OR "fibrinopeptide B" OR hemostat* OR haemostat* OR thrombocyt* OR platelet* OR d-dimer* OR "tissue factor*" OR thrombophilia OR thrombocytopenia OR pt OR "prothrombin time" OR aptt OR "activated partial thromboplastin time" OR inr OR "international normalised ratio" OR "APTT Ratio" OR "vitamin K" OR "fibrin degradation products" OR fdp OR "fibrin split products" OR fsp OR "Plasminogen activator inhibitor-1" OR pai-1 OR "Alpha-2 plasmin inhibitor" OR "antiplasmin" OR thromboelasto*)<br>4. TITLE-ABS-KEY (coagul* AND marker*)<br>5. 3 OR 4<br>6. TITLE-ABS-KEY ("atrial fibrillation" OR "AF" OR "new onset atrial fibrillation" OR new-onset-atrial-fibrillation OR "NOAF" OR "atrial flutter*" OR "supraventricular tachycardia" OR supraventricular-tachycardia OR "supraventricular arrhythmia" OR supraventricular-arrhythmia OR "SVT" OR "atrial tachycardia")<br>7. 2 OR 5<br>8. 1 AND 6 AND 7 |

This table describes the full search strategy for this study protocol by database (Medline, Cochrane Library and Scopus).

Due to translation limitations, only English language studies or studies with available translations will be considered eligible for this study. Single case reports, case series or expert opinions will also be excluded for data extraction due to limited quality of evidence. Non-full text and non-published articles will also be excluded (Table 2).

**Table 2. Inclusion/exclusion criteria.**

| Inclusion criteria | Exclusion criteria |
|---|---|
| Human studies | Animal or in vitro studies |
| Studies that report the incidence of NOAF in adult patients with sepsis* | Studies with pediatric populations (<18 years of age) |
| Studies in English or translated to English | Non-English language studies |
| Studies that include quantitative data on coagulation biomarkers and/or inflammatory biomarkers in patients with sepsis. (a full list of biomarkers is available in Table 1) | Pharmaceutical trials or articles that only discuss specific treatments (for example: betablockers) |
| | Articles that only contain expert opinions or a single case report or case series |
| | Not available as full text articles (abstract only) or non-published articles |

*Sepsis will be defined as per the third international consensus definition of sepsis [25].

Pathology Studied: Sepsis

Comorbidity Studied: NOAF

Risk Factors: 1. Markers of immune activation. 2. Factors of coagulation and fibrinolysis

Location: Critical care, Intensive care.

Table outlining the inclusion/exclusion criteria for this protocol.

### Quality assessment (risk of bias)

Potential risk of bias, will be assessed utilizing the Newcastle Ottawa Scale (NOS) for the assessment of non-randomized studies. The NOS is comprised of three domains which contribute to the overall quality score: selection of groups, comparability, and outcome (cohort studies)/ exposure (case-control). Articles are rated semi- quantitatively between zero to nine stars [26].

Randomized studies, will be assessed using the Risk of Bias 2 tool (RoB2) as recommended by Cochrane Reviews. RoB2 consists of five domains which assess for potential bias arising from: randomization, deviations from intended interventions, missing outcome data, measurement of the outcome and selection of the reported result. Studies will be classified as low risk of bias, some concerns, or a high risk of bias [27].

The Grading of Recommendations Assessment, Development and Evaluation (GRADE) approach will be used in assessing the quality of the evidence. The overall certainty in the evidence will be rated as follows: very low (1), low (2), moderate (3) and high (4) [28].

### Statistical analysis and meta-analysis

Descriptive statistical analysis will be performed using SPSS Statistics V.28.0.1.1 (IBM). Statistics will be provided regarding the coagulation and immune biomarkers in both the NOAF and non-NOAF patient populations. The mean±SEM (standard error of the mean) will be calculated for the coagulation and immune biomarkers in the NOAF and non-NOAF groups, with p values provided where possible. The Student's T-test will be performed for parametric data and for non-parametric data we will utilize the Mann-Whitney U test. A rejection threshold for our hypothesis is set at $P > 0.05$.

RevMan V5.4 (Cochrane) will be utilized for meta-analysis of the available data. When reporting subsequent outcomes in patients, odds ratios (ORs) will be calculated, with their corresponding CIs (confidence intervals). Data after extraction will be displayed in the form of forest plot diagrams. Statistical heterogeneity between the studies will be assessed using Cochran's Q test to examine the likelihood of study variance, with $I^2$ used to estimate the percentage of total variability due to heterogeneity between the included studies. For heterogeneity, <25% indicates low, 25–75% moderate and >75% high heterogeneity. In the event of significant heterogeneity, random effect models can be utilized. If there is insufficient data for quantitative analysis, a narrative synthesis will be performed to explore the existing ideas within the literature.

## Discussion

This systematic review and meta-analysis will explore the existing scientific literature regarding the association between coagulopathy and immune activation in septic critically ill patients and the development of NOAF. This study will gather and statistically analyze all available data to attempt to identify coagulation and immune biomarkers as potential risk factors for the development of NOAF. This systematic review and meta-analysis will contribute to the existing knowledge base of NOAF in sepsis, from which further research into this condition can be performed. The goal of this research is to provide clinicians with information which may allow for earlier identification of those most at risk of NOAF, in addition to guiding further management strategies for NOAF in sepsis.

### Strengths and limitations

This study has been designed using the Preferred Reporting Items for Systematic Review and Meta-Analysis Protocols (PRISMA-P) and the Meta-Analyses and Systematic Reviews of

Observational Studies (MOOSE) guidelines. Study selection will be performed by two independent reviewers with a third reviewer acting to resolve any disagreements regarding study inclusion and data extraction. Translation limitations will limit the literature to English only texts, reducing the overall data available for extraction. Any knowledge gaps will be outlined, and if there was insufficient data available for extraction and analysis, the study will act as a guide for further research into these areas. Non-randomized studies will be included which may carry a high risk of bias, however this will be discussed as a limitation in our systematic review and provide valuable information for future work.

## Supporting information

**S1 Fig. PRISMA-P checklist NOAF in sepsis.** Preferred Reporting Items for Systematic review and Meta-Analysis Protocols 2015 checklist completed for this systematic review study protocol.
(TIF)

## Acknowledgments

We wish to acknowledge our Librarian Angela Hall who was involved in conducting our literature search.

## Author Contributions

**Methodology:** Michael Downes, Brian W. Johnston.

**Writing – original draft:** Michael Downes.

**Writing – review & editing:** Michael Downes, Ingeborg D. Welters, Brian W. Johnston.

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
