## [Decision Letter · Decision Letter 0]

16 Jun 2023

PONE-D-23-10937Study Protocol: A Systematic Review and Meta-analysis Regarding the Influence of Coagulopathy and Immune activation on New Onset Atrial Fibrillation in Septic PatientsPLOS ONE

Dear Dr. Downes,

Thank you for submitting your manuscript to PLOS ONE. After careful consideration, we feel that it has merit but does not fully meet PLOS ONE’s publication criteria as it currently stands. Therefore, we invite you to submit a revised version of the manuscript that addresses the points raised during the review process.

We look forward to receiving your revised manuscript.

Kind regards,

Chiara Lazzeri

Academic Editor

PLOS ONE

Journal Requirements:

**Additional Editor Comments:**

the topic is quite interesting and the paper well written. However we suggest to better clarify the search strategy. The aims should be better elucidated considering the available evidence

Reviewers' comments:

Reviewer's Responses to Questions

**Comments to the Author**

1. Does the manuscript provide a valid rationale for the proposed study, with clearly identified and justified research questions?

Reviewer #1: Yes

2. Is the protocol technically sound and planned in a manner that will lead to a meaningful outcome and allow testing the stated hypotheses?

Reviewer #1: Partly

3. Is the methodology feasible and described in sufficient detail to allow the work to be replicable?

Reviewer #1: Yes

4. Have the authors described where all data underlying the findings will be made available when the study is complete?

Reviewer #1: Yes

5. Is the manuscript presented in an intelligible fashion and written in standard English?

Reviewer #1: Yes

6. Review Comments to the Author

You may also provide optional suggestions and comments to authors that they might find helpful in planning their study.

Reviewer #1: Thank you for the opportunity to review this protocol for a Systematic Review and Meta-analysis Regarding the Influence of Coagulopathy and Immune activation on New Onset Atrial Fibrillation in Septic Patients.

The topic is very interesting and I look forward to reading the final review. However, there are some adjustments that should be made prior to consideration of publication.

1) Has a research librarian been involved in the development of the search strategy? If so, this should be mentioned in the methods. If not then this is a significant limitation. They are most likely to be up to date with recent changes in MESH terminology and database scope.

2) I would expect to see a much clearer search strategy prior to consideration of publication, such as that outlined here: https://links.lww.com/CCM/E201. It needs to be more clear which terms are MESH, and precisely how the concepts will be combined. It should be specified which terms are being searched for in title only, or title and abstract, or title, abstract and author keywords, etc. I don’t believe it is sufficient to state “Boolean operators (OR, AND)... are to be utilised throughout the search strategy“ – these should be clarified.

3) The aim / inclusion criteria feel unfocussed. What exactly is meant by “Studies that describe AF/NOAF in relation to sepsis/ critical illness/intensive care”? Does this include any study of NOAF in the ICU? Or only studies of patients with sepsis in the ICU? Or only studies that report sepsis-related biomarkers in both NOAF and non-NOAF groups? In your aim you state “We aim to analyse the link between coagulopathy and immune activation, and the development of NOAF in septic patients”. Is this therefore a systematic review of sepsis / inflammatory / coagulation-related risk factors (biomarkers only, or vital signs as well?) for NOAF in patients admitted to an ICU? The search strategy should be tailored to these specific criteria.

4) The authors may which to re-word their approach to the I2 statistic “with I2 used to interpret the percentage of variance across the studies.” – more accurately it would estimate the percentage of total variability due to between-study heterogeneity. I would advise caution when using the I2 statistic to categorise the amount of heterogeneity (see https://doi.org/10.1002/jrsm.1230 for more details).

Minor

Introduction: The second line is a little jarring. It introduces both the concept of “new-onset”, and the higher incidence in the subgroup of patients with sepsis at the same time, and starts the sentence with “in particular” which doesn’t really work.

The manuscript would benefit from a thorough review from one of the senior authors to tidy up some of the language. Examples below:

• Introduction line 7: I’d replace “inciting” with “conferring”

• Aim: You have “hypothesized” with a z, but “analyse” with an s. Please standardise / standardize. Suggest a z for this international journal.

• Protocol design: You state “in accordance with the Preferred Reporting Items for Systematic Reviews and Meta-Analyses guideline (PRISMA-P)” – you have missed the word “Protocols” – same in “strengths and limitations”.

• Data sourcing: phrases like “Utilising our research question…” are unnecessary in this context

• Clear language: Suggest replacing most instances of “shall” with “will”, e.g. in “We shall assess risk of bias…”

• Clear language: In the abstract, I think “management and treatment strategies for NOAF” could be simplified to “management strategies for NOAF”.

• “Septic patients” is a little labelling. I would encourage a change to “patients with sepsis” throughout the manuscript.

7. PLOS authors have the option to publish the peer review history of their article (what does this mean?). If published, this will include your full peer review and any attached files.

Reviewer #1: No

---

## [Author Response · Author response to Decision Letter 0]

8 Aug 2023

) Has a research librarian been involved in the development of the search strategy? If so, this should be mentioned in the methods. If not then this is a significant limitation. They are most likely to be up to date with recent changes in MESH terminology and database scope.

A research librarian was involved in the development of our search strategy and retrieving all relevant articles. We have updated our manuscript to reflect this and have also include the librarian in our acknowledgements.

2) I would expect to see a much clearer search strategy prior to consideration of publication, such as that outlined here: https://links.lww.com/CCM/E201. It needs to be more clear which terms are MESH, and precisely how the concepts will be combined. It should be specified which terms are being searched for in title only, or title and abstract, or title, abstract and author keywords, etc. I don’t believe it is sufficient to state “Boolean operators (OR, AND)... are to be utilised throughout the search strategy“ – these should be clarified.

We thank the reviewers for their suggestion and have included our complete search strategy within Table 1. We have included our search strategy for each database searched. 

3) The aim / inclusion criteria feel unfocussed. What exactly is meant by “Studies that describe AF/NOAF in relation to sepsis/ critical illness/intensive care”? Does this include any study of NOAF in the ICU? Or only studies of patients with sepsis in the ICU? Or only studies that report sepsis-related biomarkers in both NOAF and non-NOAF groups? In your aim you state “We aim to analyse the link between coagulopathy and immune activation, and the development of NOAF in septic patients”. Is this therefore a systematic review of sepsis / inflammatory / coagulation-related risk factors (biomarkers only, or vital signs as well?) for NOAF in patients admitted to an ICU? The search strategy should be tailored to these specific criteria.

We have significantly reworded these sections of the manuscript to ensure clarity. Our aim is to assess if individual coagulation or inflammatory biomarkers can be utilised to identify patients with sepsis that are at risk of developing NOAF. We wish to retrieve articles that report biomarkers in patients admitted with sepsis who develop and do not develop NOAF. We are limiting our review to biomarkers only and therefore not including physio markers such as vital signs. Physio markers as risk factors for NOAF has been extensively reported elsewhere and will therefore not be the focus of this review. We will use the most recent sepsis 3 definition to define our sepsis population. We anticipate that some studies will predate the sepsis 3 consensus definition (2016) however, the definition is broader than previous definitions so we are confident that all relevant papers will be included in this definition. 

4) The authors may which to re-word their approach to the I2 statistic “with I2 used to interpret the percentage of variance across the studies.” – more accurately it would estimate the percentage of total variability due to between-study heterogeneity. I would advise caution when using the I2 statistic to categorise the amount of heterogeneity (see https://doi.org/10.1002/jrsm.1230 for more details).

The I2 statistic has been reworded to more accurately represent its use, see updated manuscript.

Minor

Introduction: The second line is a little jarring. It introduces both the concept of “new-onset”, and the higher incidence in the subgroup of patients with sepsis at the same time, and starts the sentence with “in particular” which doesn’t really work.

We thank the reviewers for their comments. We have re-written the introduction to improve clarity of the manuscript. We have also made it clear that new onset atrial fibrillation is a subgroup of atrial fibrillation that is common in sepsis. 

The manuscript would benefit from a thorough review from one of the senior authors to tidy up some of the language. Examples below:

• Introduction line 7: I’d replace “inciting” with “conferring”

Inciting has been replaced with conferring as requested and the manuscript has been re-reviewed by a senior author for clarity. 

• Aim: You have “hypothesized” with a z, but “analyse” with an s. Please standardise / standardize. Suggest a z for this international journal.

Language has been converted to standard American English, see manuscript. 

• Protocol design: You state “in accordance with the Preferred Reporting Items for Systematic Reviews and Meta-Analyses guideline (PRISMA-P)” – you have missed the word “Protocols” – same in “strengths and limitations”.

Protocols has been added to all mentions of PRISMA-P.

• Data sourcing: phrases like “Utilising our research question…” are unnecessary in this context.

Unnecessary phrases such as “Utilising our research question…” have been removed.

• Clear language: Suggest replacing most instances of “shall” with “will”, e.g. in “We shall assess risk of bias…”

Instances of shall have been replaced with will for clarity.

• Clear language: In the abstract, I think “management and treatment strategies for NOAF” could be simplified to “management strategies for NOAF”.

“management and treatment strategies for NOAF” has been simplified to “management strategies for NOAF”.

• “Septic patients” is a little labelling. I would encourage a change to “patients with sepsis” throughout the manuscript.

Patients with sepsis is now used throughout the document.

---

## [Editor Report · Decision Letter 1]

21 Aug 2023

Study Protocol: A Systematic Review and Meta-analysis Regarding the Influence of Coagulopathy and Immune activation on New Onset Atrial Fibrillation in Septic Patients

PONE-D-23-10937R1

Dear Dr. Downes,

We’re pleased to inform you that your manuscript has been judged scientifically suitable for publication and will be formally accepted for publication once it meets all outstanding technical requirements.

Kind regards,

Chiara Lazzeri

Academic Editor

PLOS ONE
---

## [Editor Report · Acceptance letter]

29 Aug 2023

PONE-D-23-10937R1 

Study protocol: a systematic review and meta-analysis regarding the influence of coagulopathy and immune activation on new onset atrial fibrillation in patients with sepsis 

Dear Dr. Downes:

I'm pleased to inform you that your manuscript has been deemed suitable for publication in PLOS ONE. Congratulations! Your manuscript is now with our production department. 

Kind regards, 

on behalf of

Dr. Chiara Lazzeri 

Academic Editor

PLOS ONE